# Generalization vs Specialization under Concept Shift

**Alex Nguyen**
Princeton University

**David J. Schwab**[*]
CUNY Graduate Center

**Vudtiwat Ngampruetikorn**[*]
University of Sydney

## Abstract

Machine learning models are often brittle under distribution shift, i.e., when data distributions at test time differ from those during training. Understanding this failure mode is central to identifying and mitigating safety risks of mass adoption of machine learning. Here we analyze ridge regression under concept shift—a form of distribution shift in which the input-label relationship changes at test time. We derive an exact expression for prediction risk in the thermodynamic limit. Our results reveal nontrivial effects of concept shift on generalization performance, including a phase transition between weak and strong concept shift regimes and nonmonotonic data dependence of test performance even when double descent is absent. Our theoretical results are in good agreement with experiments based on transformers pretrained to solve linear regression; under concept shift, too long context length can be detrimental to generalization performance of next token prediction. Finally, experiments on MNIST and FashionMNIST further validate our theoretical predictions, suggesting these phenomena represent a fundamental aspect of learning under distribution shift.

## 1 Distribution shift

It is unsurprising that a model trained on one distribution does not perform well when applied to data from a different distribution. Yet, this out-of-distribution setting is relevant to many practical applications from scientific research [1, 2] to medicine and healthcare [3–6]. A quantitative understanding of out-of-distribution generalization is key to developing safe and robust machine learning techniques. A model that generalizes to arbitrary distribution shifts of course does not exist. The generalization scope of a model, however, needs not be limited to the training data distribution. A question then arises as to how much a model's scope extends beyond its training distribution.

Answering this question requires assumptions on the test distribution. For example, covariate shift, a well-studied setting for distribution shift, assumes a fixed input-label relationship while allowing changes in the input distribution (see, e.g., Refs [7–11]).

We consider *concept shift*—a relatively less-studied setting, in which the input-label relationship becomes different at test time[1] [13, 14], see Fig 1. While many works have studied how to detect and mitigate concept shift [15, 16], characterizing how concept shift affects generalization behavior in neural networks has remained underexplored. In our work, we formulate a minimal model which enables continuous modulation of the input-label function, based on high dimensional ridge regression—a solvable setting that has helped develop intuitions for some of the most interesting phenomena of modern machine learning (see, e.g., Refs [17–25]). We derive an analytical expression for prediction risk under concept shift in the high-dimensional limit and demonstrate that its behavior can change from monotonically decreasing with data for the in-distribution case to monotonically

---

[*]DJS and VN contributed to this work equally.

[1]Concept shift or concept drift is sometimes defined to be equivalent to distribution shift [12]. Here, we adopt a narrower definition in which concept shift describes only the change in the input-label relationship [13, 14].

39th Conference on Neural Information Processing Systems (NeurIPS 2025).

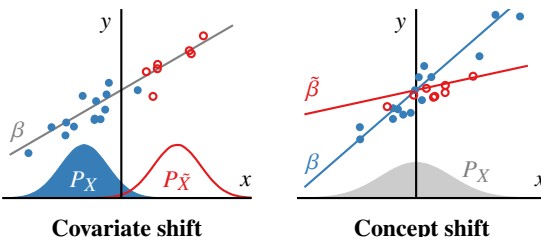

Figure 1: **Two flavors of distribution shift.** Distribution shift describes the scenarios in which the joint input-label distribution of training data $\{(x_1, y_1), (x_2, y_2), \dots\}$ (filled circles) differs from that of test data $\{(\tilde{x}_1, \tilde{y}_1), (\tilde{x}_2, \tilde{y}_2), \dots\}$ (empty circles). *Covariate shift* (left) assumes a fixed input-label relationship but the input distribution differs at test time, i.e., $P_{Y|X} = P_{\tilde{Y}|\tilde{X}}$ but $P_X \neq P_{\tilde{X}}$. For linear regression, this condition means that the regression coefficient $\beta$ is unchanged. *Concept shift* (right) allows the input-label function to change at test time, which corresponds to a shift in regression coefficient $\beta \neq \tilde{\beta}$ in the linear regression setting. See §2.

increasing, and to nonmonotonic, depending on the degree of concept shift and the properties of the input distribution.

To differentiate these effects from double descent phenomena which can also cause nonmonotonic data dependence of prediction risk [26, 27], we focus on optimally tuned ridge regression which completely suppresses the risk divergence at the interpolation threshold and for which in-distribution prediction risk decreases monotonically with more data [17, 18, 28]. The nonmonotonic behavior we observe is also distinct from effects due to model misspecification [18]. While misspecification—a property of a model relative to the true data-generating process—can produce similar generalization behavior, distribution shift represents a fundamentally different setting in which the data-generating process changes at test time. Our work isolates and characterizes the effects of concept shift, a type of distribution shift, in correctly specified, optimally regularized models.

Our theoretical results in the thermodynamic limit agree well with experiments, based on transformers trained to solve finite-dimensional regression using in-context examples. We show that more in-context examples help improve model performance when concept shift is weak, but can lead to overspecialization for strong concept shift. Finally, we illustrate similar qualitative changes in generalization behavior in classification problems, using MNIST and FashionMNIST as examples. Our work contributes a new theoretical framework for analyzing concept shift that complements an extensive body of work on concept shift detection (see, e.g., Ref [29] for a recent review).

Our main contributions are:

1. We develop an analytically solvable framework that isolates the effects of concept shift—an important yet often overlooked mode of distribution shift.
2. We explain precisely why and how more training data can hurt generalization performance under concept shift, illustrating its effects in several specific settings, including coefficient shrinking, rotation, and feature robustness.
3. We identify and characterize a sharp transition, separating weak and strong concept shift regimes.
4. We show that feature anisotropy creates qualitatively different patterns of risk nonmonotonicity depending on whether concept shift affects high or low-variance features.

Our experiments on transformers and simple classification tasks are in good qualitative agreement with our theoretical findings, hinting at universal behavior that generalizes beyond our relatively simple theoretical settings.

## 2 Regression setting

**Data.** The training data consists of $N$ iid input-response pairs $\{(x_1, y_1), \dots, (x_N, y_N)\}$. The input $x \in \mathbb{R}^P$ is a vector of Gaussian features and the response $y \in \mathbb{R}$ is a noisy linear projection of $x$, i.e.,

$$y = \beta^{\mathsf{T}} x + \xi \quad \text{with} \quad (x, \xi) \sim \mathcal{N}(\cdot, \Sigma) \times \mathcal{N}(\cdot, \sigma_\xi^2), \tag{1}$$

where $\beta \in \mathbb{R}^P$ denotes the coefficient vector, $\xi \in \mathbb{R}$ Gaussian noise with variance $\sigma_\xi^2$, and $\Sigma \in \mathbb{R}^{P \times P}$ the covariance matrix. Similarly, a test data point is an input-response pair $(\tilde{x}, \tilde{y})$, drawn from the same process as the training data, Eq (1), but with a generally different set of parameters—that is,

$$\text{Training data:} \quad \begin{bmatrix} x \\ y \end{bmatrix} \sim \mathcal{N}\left( \cdot, \begin{bmatrix} \Sigma & \Sigma\beta \\ \beta^\mathsf{T}\Sigma & \beta^\mathsf{T}\Sigma\beta + \sigma_\xi^2 \end{bmatrix} \right)$$

$$\text{Test data:} \quad \begin{bmatrix} \tilde{x} \\ \tilde{y} \end{bmatrix} \sim \mathcal{N}\left( \cdot, \begin{bmatrix} \tilde{\Sigma} & \tilde{\Sigma}\tilde{\beta} \\ \tilde{\beta}^\mathsf{T}\tilde{\Sigma} & \tilde{\beta}^\mathsf{T}\tilde{\Sigma}\tilde{\beta} + \tilde{\sigma}_\xi^2 \end{bmatrix} \right), \tag{2}$$

where in general $\Sigma \neq \tilde{\Sigma}$, $\beta \neq \tilde{\beta}$ and $\sigma_\xi^2 \neq \tilde{\sigma}_\xi^2$. Here we also define signal-to-noise ratio $\text{SNR} \equiv \beta^\mathsf{T}\Sigma\beta/\sigma_\xi^2$

**Model.** We consider ridge regression in which the predicted response to an input $x$ reads $\hat{y}(x; X, Y) = x \cdot \hat{\beta}_\lambda(X, Y)$, with the coefficient vector resulting from minimizing $L_2$-regularized mean squared error,

$$\hat{\beta}_\lambda(X, Y) \equiv \arg\min_{b \in \mathbb{R}^P} \frac{1}{N} \|Y - X^\mathsf{T}b\|^2 + \lambda\|b\|^2 = (XX^\mathsf{T} + \lambda N I_P)^{-1} XY. \tag{3}$$

Here $\lambda > 0$ controls the regularization strength, and $Y = (y_1, \ldots, y_N)^\mathsf{T} \in \mathbb{R}^N$ and $X = (x_1, \ldots, x_N)^\mathsf{T} \in \mathbb{R}^{P \times N}$ denote the training data.

**Risk.** We measure generalization performance with prediction risk,[2]

$$R(X) \equiv \mathbf{E}\left[ \|\hat{y}(\tilde{x}; X, Y) - \mathbf{E}(\tilde{y} \mid \tilde{x})\|^2 \mid X \right] = B(X) + V(X), \tag{4}$$

where the last equality denotes the standard bias-variance decomposition with

$$B(X) \equiv \mathbf{E}\left[ \|\mathbf{E}(\hat{y}(\tilde{x}; X, Y) \mid X, \tilde{x}) - \mathbf{E}(\tilde{y} \mid \tilde{x})\|^2 \mid X \right]$$

$$V(X) \equiv \mathbf{E}\left[ \|\hat{y}(\tilde{x}; X, Y) - \mathbf{E}(\hat{y}(\tilde{x}; X, Y) \mid X, \tilde{x})\|^2 \mid X \right].$$

Substituting the predictor from ridge regression into the above equations yields

$$B(X) = \left( \frac{\Psi}{\Psi + \lambda I_P}\beta - \tilde{\beta} \right)^\mathsf{T} \tilde{\Sigma} \left( \frac{\Psi}{\Psi + \lambda I_P}\beta - \tilde{\beta} \right) \quad \text{and} \quad V(X) = \sigma_\xi^2 \frac{1}{N} \operatorname{Tr}\left( \tilde{\Sigma} \frac{\Psi}{(\Psi + \lambda I_P)^2} \right), \tag{5}$$

where $\Psi \equiv XX^\mathsf{T}/N$ is the empirical covariance matrix.

It is instructive to consider the idealized limits of $N = 0$ and $N \to \infty$. First, when $N = 0$, inductive biases (e.g., from model initialization and regularization) dominate. For ridge regression, Eq (3), all model parameters vanish, $\hat{\beta}_\lambda = 0$, and the resulting predictor outputs zero regardless of the input, i.e., $\hat{y}(x) = 0$ for any $x$. As a result, $R_{N=0}(X) = \mathbf{E}[\mathbf{E}(\tilde{y} \mid \tilde{x})^2] = \tilde{\beta}^\mathsf{T}\tilde{\Sigma}\tilde{\beta}$, see Eq (4). Second, when $N \to \infty$, the empirical covariance matrix approaches the true covariance matrix $\Psi \to \Sigma$ and, taking the limit $\lambda \to 0^+$, we obtain $R_{N \to \infty}(X) = (\beta - \tilde{\beta})^\mathsf{T}\tilde{\Sigma}(\beta - \tilde{\beta})$.[3] When $\tilde{\beta} = \beta$, we see that $R_{N=0}(X) = \beta^\mathsf{T}\tilde{\Sigma}\beta > 0$ whereas $R_{N \to \infty}(X) = 0$ (see also Fig 2). That is, infinite data is better than no data, as expected.

This intuitive picture breaks down under concept shift, $\tilde{\beta} \neq \beta$. Consider, for example, $\tilde{\beta} = 0$ which indicates that none of the features predicts the response at test time. In this case, $R_{N=0}(X) = 0$ and $R_{N \to \infty}(X) = \beta^\mathsf{T}\tilde{\Sigma}\beta > 0$ (see also Fig 2); that is, even *infinitely more* data decreases test performance in the presence of concept shift.

## 3 High dimensional limit

To better understand this counterintuitive phenomenon in the context of high dimensional learning, we focus on concept shift without covariate shift, i.e., $\tilde{\beta} \neq \beta$ and $\tilde{\Sigma} = \Sigma$, and take the thermodynamic limit $N, P \to \infty$ and $P/N \to \gamma \in (0, \infty)$. In this limit, prediction risk becomes deterministic $R(X) \to \mathcal{R}$ with the bias and variance contributions given by (see Appendix for derivation; see also Ref [30]),

$$B(X) \to \mathcal{B} = \lambda^2 \nu'(-\lambda) \frac{\beta^\mathsf{T}\Sigma\hat{G}_\Sigma^2(-\lambda)\beta}{\frac{1}{P}\operatorname{Tr}[\Sigma\hat{G}_\Sigma^2(-\lambda)]} - 2\lambda\beta^\mathsf{T}\Sigma\hat{G}_\Sigma(-\lambda)(\beta - \tilde{\beta}) + (\beta - \tilde{\beta})^\mathsf{T}\Sigma(\beta - \tilde{\beta}) \tag{6}$$

$$V(X) \to \mathcal{V} = \sigma_\xi^2 \gamma \left[ \nu(-\lambda) - \lambda\nu'(-\lambda) \right], \tag{7}$$

---

[2]We follow the convention in Ref [18] where the risk is the expected squared error between the predictor and the mean of the test-time conditional distribution, effectively subtracting the irreducible noise variance.

[3]As $N \to \infty$ at fixed $P$, the variance term vanishes, $\mathcal{V}(X) \sim O(N^{-1})$, and the optimal regularization is $\lambda^* = 0$.

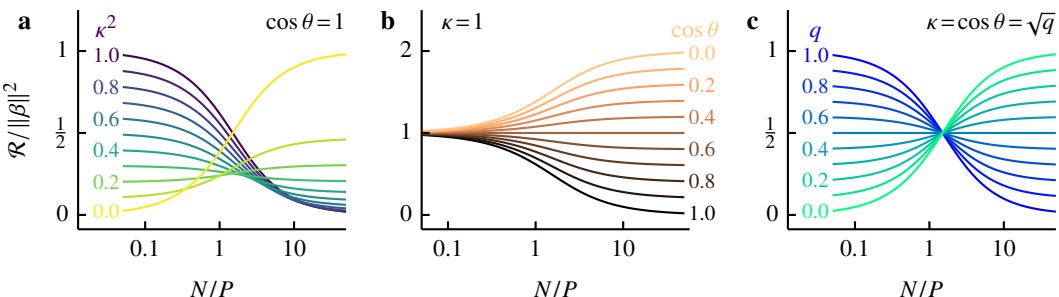

Figure 2: **More data hurts performance when concept shift is strong.** We depict the data dependence of asymptotic prediction risk for isotropic features, Eq (9), under three concept shift settings of varying degree, parametrized by coefficient alignment $\cos\theta = \beta \cdot \tilde{\beta} / \|\beta\| \|\tilde{\beta}\|$ and scaling factor $\kappa = \|\tilde{\beta}\| / \|\beta\|$ (see legend). **a** *Shrinking coefficients:* $\tilde{\beta} = \kappa\beta$. **b** *Rotating coefficients:* $\|\tilde{\beta}\| = \|\beta\|$ but $\theta$ varies. **c** *Mixture of robust and nonrobust features:* $\tilde{\beta}_i = \beta_i$ if feature $i$ is robust, otherwise $\tilde{\beta}_i = 0$. This setting is parametrized by $q = \kappa^2 = \cos^2\theta$. We set SNR = 1 in a-c, and the regularization is in-distribution optimal $\lambda = \gamma / \text{SNR}$.

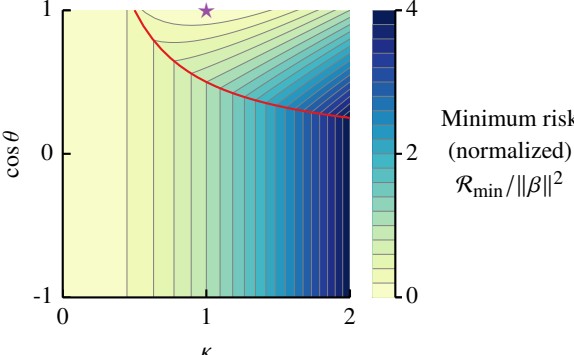

Figure 3: **Concept shift induces a phase transition in generalization behavior.** We depict minimum prediction risk $\mathcal{R}_{\min}$ of optimally tuned ridge regression ($\lambda = \gamma / \text{SNR}$) as $\kappa$ and $\cos\theta$ vary, see Eq (10) for definitions. No concept shift corresponds to $\kappa = \cos\theta = 1$ (star). The thick curve, $\kappa\cos\theta = 1/2$, separates the weak and strong concept shift regimes. More training data improves generalization only when concept shift is weak, $\kappa\cos\theta > 1/2$. Outside this region, *any* data hurts generalization.

where we define $\hat{G}_\Sigma(z) \equiv (m(z)\Sigma - zI_P)^{-1}$, $m(z) \equiv (1 + \gamma\nu(z))^{-1}$ and $\nu(z)$ is the unique solution of the self-consistent equation,

$$\nu(z) = \frac{1}{P}\text{Tr}[\Sigma\hat{G}_\Sigma(z)] \quad \text{with} \quad \nu(z) \in \mathbb{C}^+. \tag{8}$$

We note that concept shift enters prediction risk only through the last two terms of the bias, Eq (6), whereas the variance, Eq (7), is completely unaffected.

## 3.1 Isotropic features

When $\Sigma = I_P$, the bias contribution to prediction risk reads (see Appendix for a closed-form expression for $\nu(z)$)

$$\mathcal{B} = \|\beta\|^2 \left[ \lambda^2\nu'(-\lambda) - \frac{2\lambda(1 + \gamma\nu(-\lambda))}{1 + \lambda(1 + \gamma\nu(-\lambda))}(1 - \kappa\cos\theta) + 1 - 2\kappa\cos\theta + \kappa^2 \right], \tag{9}$$

where we quantify concept shift via two parameters

$$\begin{aligned} \textit{Coefficient alignment:} \quad &\cos\theta \equiv \frac{\beta \cdot \tilde{\beta}}{\|\beta\|\|\tilde{\beta}\|} \\ \textit{Scaling factor:} \quad &\kappa \equiv \|\tilde{\beta}\| / \|\beta\|. \end{aligned} \tag{10}$$

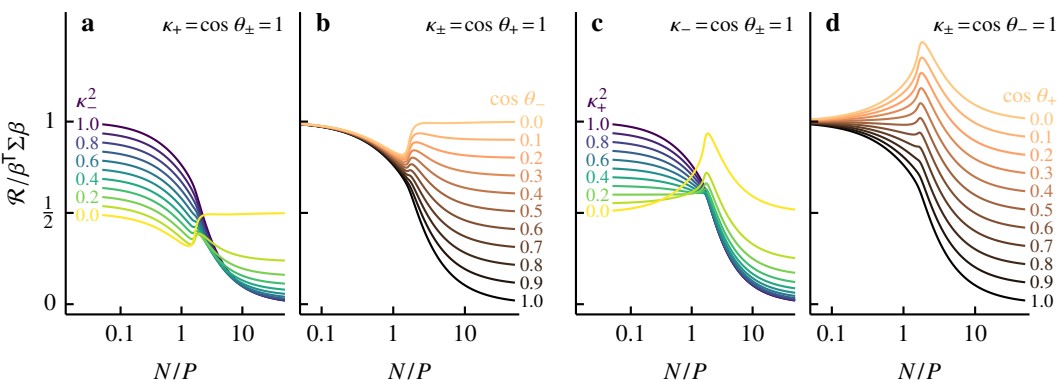

Figure 4: **Anisotropic features can lead to risk nonmonotonicity.** We illustrate prediction risk for the two-scale model, Eq (11), with aspect ratio $s_-/s_+ = 0.1$, spectral weights $\rho_+ = \rho_- = 1/2$ and signal fraction $\pi_+ = \pi_- = 1/2$. Concept shift is parametrized by coefficient alignments $\cos\theta_\pm = \beta_\pm \cdot \tilde\beta_\pm / \|\beta_\pm\|\|\tilde\beta_\pm\|$ and scaling factors $\kappa_\pm = \|\tilde\beta_\pm\|/\|\beta_\pm\|$ where the subscripts indicate the variance $s_\pm$ of the affected features. We consider the settings in which concept shift affects either low or high-variance features and either alignment or scale; that is, we vary only one out of four parameters, $\theta_\pm$ and $\kappa_\pm$, at a time (see legend). **a** Shrinking coefficients of low-variance features. **b** Rotating coefficients of low-variance features. **c** and **d** Same as Panels a and b, but concept shift affects high-variance features via $\kappa_+$ and $\cos\theta_+$, respectively. Here SNR = 1 and regularization is in-distribution optimal.

Figure 2 depicts thermodynamic-limit prediction risk for isotropic features under concept shift. We focus on in-distribution optimal ridge regression which corresponds to setting $\lambda = \gamma/\text{SNR}$, ensuring that double descent is absent (see, e.g., Refs [17, 18]). This choice is motivated by its practical relevance (as it can be approximated via cross-validation) and because it allows us to isolate the effects of concept shift from the confounding non-monotonicity of double descent.

In Fig 2a, we consider the effects of shrinking coefficients—the coefficient vector becomes smaller at test time without changing direction, $\theta = 0$ and $\kappa \leq 1$. When $\kappa = 1$, concept shift is absent and prediction risk monotonically decreases with more training data, as expected for optimally-tuned ridge regression. As $\kappa$ decreases and the features become less predictive of the response at test time, prediction risk starts to increase with training sample size. This transition occurs at $\kappa = 1/2$, at which $\mathcal{R}/\|\beta\|^2 = 1/4$.

In Fig 2b, we observe a similar crossover when the magnitude of the coefficient vector is fixed but its direction changes at test time, $\kappa = 1$ and $\theta \geq 0$. Examples of this concept shift setting include the case where some features have the opposite effects at test time, described by $\tilde\beta_i = -\beta_i$ for the affected coefficient. Here we see that more data hurts when $\cos\theta < 1/2$ (or equivalently $\theta > \pi/3$).

In Fig 2c, we consider a mixture of robust and nonrobust features with $\tilde\beta_i = \beta_i$ if feature $i$ is robust, otherwise $\tilde\beta_i = 0$. That is, the robust features have the same effects at test time whereas the nonrobust ones become uninformative of the response variables. In this case, we have $\kappa = \cos\theta = \sqrt{q}$ where $0 \leq q \leq 1$ denotes the fraction of robust features. We see again that adequately strong concept shift changes the data-dependent behavior of generalization properties from improving to worsening with more training data.

Indeed we can make a more general statement about the transition between weak and strong concept shift. First, we observe that for in-distribution optimal ridge regression, $\lambda = \gamma/\text{SNR}$, the thermodynamic-limit risk is always monotonic in $\gamma$. To determine whether more data improve generalization, we only need to compare the limits of no data and infinite data: $\mathcal{R}_{N=0} = \kappa^2\|\beta\|^2$ and $\mathcal{R}_{N\to\infty} = (1 - 2\kappa\cos\theta + \kappa^2)\|\beta\|^2$. It follows immediately that two qualitatively distinct regimes exist. When $\kappa\cos\theta > 1/2$, concept shift is weak and more data improves generalization. When $\kappa\cos\theta < 1/2$, concept shift is strong and more data hurts, see Fig 3. In the thermodynamic limit, this sharp boundary manifests as a nonanalyticity in the minimal prediction risk, analogous to a phase transition in statistical mechanics. When $\kappa\cos\theta = 1/2$, prediction risk becomes completely

independent of training sample size—i.e., $\mathcal{R} = \mathcal{R}_{N=0} = \mathcal{R}_{N\to\infty} = \kappa^2\|\beta\|^2$. Quite remarkably, this analysis does not depend on SNR.[4]

## 3.2 Anisotropic features

To study the effects of anisotropy, we consider a two-scale model in which the spectral density of the covariance matrix is a mixture of two point masses at $s_-$ and $s_+$ with weights $\rho_-$ and $\rho_+ = 1-\rho_-$, respectively. Without loss of generality, we let $s_+ \geq s_-$ throughout this section. These two modes define subspaces into which we decompose the coefficients, $\beta = \beta_- + \beta_+$ with $\Sigma\beta_\pm = s_\pm\beta_\pm$, and similarly for $\tilde{\beta}$. We write down the bias contribution to prediction risk

$$\mathcal{B} = \beta^\mathsf{T}\Sigma\beta \sum_{\tau\in\pm} \pi_\tau \left[ \lambda^2 \frac{\nu'(-\lambda)g_\tau^2(-\lambda)}{\sum_{\nu\in\pm}\rho_\nu s_\nu g_\nu^2(-\lambda)} - 2\lambda g_\tau(-\lambda)(1 - \kappa_\tau\cos\theta_\tau) + 1 - 2\kappa_\tau\cos\theta_\tau + \kappa_\tau^2 \right], \quad (11)$$

where $g_\pm(z) \equiv (m(z)s_\pm - z)^{-1}$ and $\pi_\pm \equiv \beta_\pm^\mathsf{T}\Sigma\beta_\pm/\beta^\mathsf{T}\Sigma\beta$ denotes the signal fraction at each scale. Similarly to the isotropic case, we quantify concept shift using coefficient alignments $\cos\theta_\pm \equiv \beta_\pm\cdot\tilde{\beta}_\pm/\|\beta_\pm\|\|\tilde{\beta}_\pm\|$ and scaling factors $\kappa_\pm \equiv \|\tilde{\beta}_\pm\|/\|\beta_\pm\|$, both of which now depend also on variance.

Figure 4 illustrates prediction risk under concept shift for two-scale covariates. We numerically tune the ridge regularization strength $\lambda$ such that the in-distribution risk is minimized and the divergences associated with multiple descent phenomena are completely suppressed. We consider the cases where concept shift affects only low-variance features via either $\kappa_-$ or $\cos\theta_-$ (Fig 4a and b), or only high-variance ones via either $\kappa_+$ or $\cos\theta_+$ (Fig 4c and d). In all cases, we see that test performance develops nonmonotonic data dependence as test data deviates from in-distribution settings; however, its behavior is qualitatively distinct for low and high-variance concept shift. To isolate the effects of covariate statistics, we set the signal fraction to $\pi_+ = \pi_- = \frac{1}{2}$ so that low and high-variance features carry the same signal strength.

In Fig 4a and b, we depict the effects of concept shift on low-variance features. Panel a corresponds to the case where the low-variance coefficients $\beta_-$ shrink at test time, $\kappa_- \leq 1$, thus suppressing the signal associated with low-variance features. Similarly to the isotropic case (Fig 2a), prediction risk decreases with $N$ for weak concept shift. However, as $\|\beta_-\|$ becomes smaller, we see that generalization performance exhibits nonmonotonic behavior; too much training data can worsen generalization. Panel b turns to the case in which $\beta_-$ rotates at test time, $\cos\theta_- \leq 1$ (cf. Fig 2b). We observe a similar crossover in the data dependence of prediction risk as concept shift becomes stronger. While generalization improves with more data in the low-data limit, this improvement continues monotonically only for adequately weak concept shifts.

In Fig 4c and d, concept shift affects only high-variance features via shrinking coefficients (panel a) and rotated coefficients (panel b). We see again that strong concept shift leads to nonmonotonic data dependence of prediction risk. However, strong concept shift on high-variance features results in risk maximum at intermediate training data size $N$. This behavior contrasts sharply with low-variance concept shift, depicted in Fig 4a and b.

Indeed we could have anticipated the intriguing differences between concept shift affecting low and high-variance features. The data dependence in Fig 4 results from the fact that it takes more data to learn low-variance features and their effects. At low $N$, high-variance features dominate prediction risk; more data hurts when these features do not adequately predict the response at test time, Fig 4c and d. On the other hand, low-variance features affect test performance only when the training sample size is large enough to influence the model. Sufficiently strong concept shift on these features thus results in detrimental effects of more data at larger $N$ (compared to high-variance concept shift), Fig 4a and b. We emphasize that the nonmonotonic data dependence of prediction risk due to concept shift is unrelated to double descent phenomena (which describe risk nonmonotonicity in suboptimally tuned models).

# 4 Transformer experiments

To test the applicability of our theoretical framework to realistic scenarios, we train transformers to perform in-context learning (ICL) of (noisy) linear functions, which were argued to perform optimal

---

[4]SNR controls how prediction risk depends on training data size $N$ (Fig 2), but not the transition between weak and strong concept shift regimes (Fig 3).

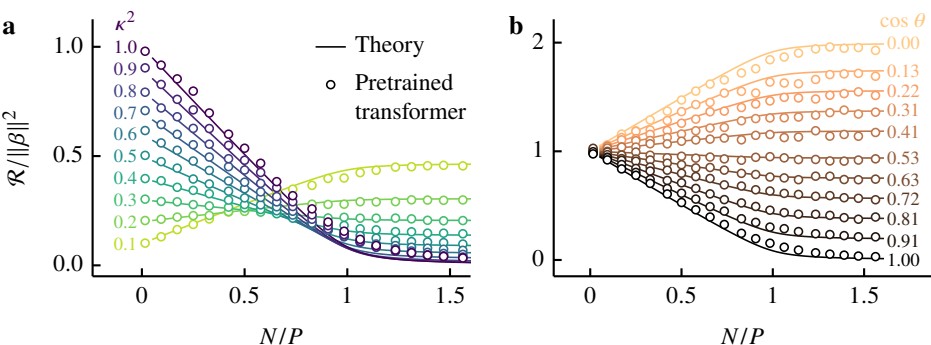

Figure 5: **Longer context can hurt in-context learning performance.** We depict prediction risk of transformers (circles), trained to solve linear regression tasks using in-context examples (see 4 for details). We compare in-context regression We consider two concept shift settings. **a** Coefficient shrinking parametrized by $\kappa$ (see legend; cf. Fig 2a). **b** Coefficient rotation parametrized by $\theta$ (see legend; cf. Fig 2b). We compare the asymptotic prediction risk of transformers on isotropic data with the predictions from our theory (Section 3) under two concept shift settings. Circles depict MSE loss attained by the transformer whereas lines depict the optimally regularized asymptotic limit. We observe strong agreement between experimental results and thermodynamic limits. Here, SNR=128.

ridge regression [31–33]. As many-shot prompting, enabled by recently expanded context windows, has shown promise in improving performance [34], it is timely to investigate how distribution shift affects whether longer context is beneficial. These experiments serve as a validation of our theory, confirming that the key phenomena derived from our analytical model persist in a more complex, data-driven setting.

In our experiment, a transformer takes as its input a series of points $(x_i, y_i)$ on an unknown function $y_i = f(x_i)$ for $i = 1, 2, \ldots, n - 1$, terminating with a 'query' $x_n$ whose function value $y_n$ is the prediction target. The model is trained to minimize the mean square loss $\mathcal{L} = \mathbb{E}[(y_n - \hat{y}_n)^2]$. This setup has proved valuable in developing intuitions about ICL [31–33, 35–38]. Here, we focus on noisy linear functions—$f_\beta(x) = \beta^\mathsf{T} x + \xi$ with $\xi$ denoting the noise term—and investigate the generalization properties of the learned ICL solution, implemented by a transformer, under concept shift at test time.

We consider linear regression in $P = 32$ dimensions. The training tasks, parametrized by $\beta$, are drawn iid from a standard normal distribution and kept fixed during training. We generate a total of $2^{20}$ training tasks, which are sufficient to elicit general-purpose ICL [33, 37]. An input sequence is drawn from $y = \beta^\mathsf{T} x + \xi$ with $(x, \xi) \sim \mathcal{N}(0, I_P) \times \mathcal{N}(0, \sigma^2)$. We choose $\sigma^2 = 0.5$. The input-response pairs are newly generated each time the model takes an input sequence. We adopt the nanoGPT architecture [39] with eight layers, an embedding dimension of 128, learnable position embeddings, and causal masking. The model is trained to minimize the next token mean squared error (MSE) using Adam with a learning rate of 0.0001. At test time, we sample 10,000 new tasks and compute the in-distribution prediction risk simply as the MSE of the transformer on test tasks. To compute risk under concept shift, the transformer is presented with a context, $(x_1, y_1, \ldots, x_{n-1}, y_{n-1}, \tilde{x})$, in which $y_i = \beta^\mathsf{T} x_i + \xi$. But the query $\tilde{x}$ is related to the final prediction target $\tilde{y}$ via a linear function $\tilde{y} = \tilde{\beta}^\mathsf{T} \tilde{x}$ with $\tilde{\beta} \neq \beta$ in general.

Figure 5 depicts ICL prediction risk for linear regression under concept shift as a function of in-context sample size $N$. We parametrize the degree of concept shift using the scaling factor and cosine similarity, described in Eq (10). We consider two specific settings: in panel a, $\kappa \leq 1$ and $\cos\theta = 1$, and in panel b, $\kappa = 1$ and $\cos\theta \leq 1$ (cf. Fig 2a and b). In both cases, we compare ICL prediction risk (symbols) with our thermodynamic-limit theory (lines). We see that they are in good quantitative agreement. In particular, the context-length dependence ICL prediction risk changes from improving to worsening with longer context as concept shift becomes more severe.

To further investigate how transformers handle concept shift with anisotropic features, we conducted experiments comparing models trained on isotropic versus anisotropic data. Figure 6 illustrates how feature anisotropy influences the generalization behavior of transformer-based in-context regression under varying degrees of concept shift. When exposed to two-scale anisotropic data (see §3.2) where

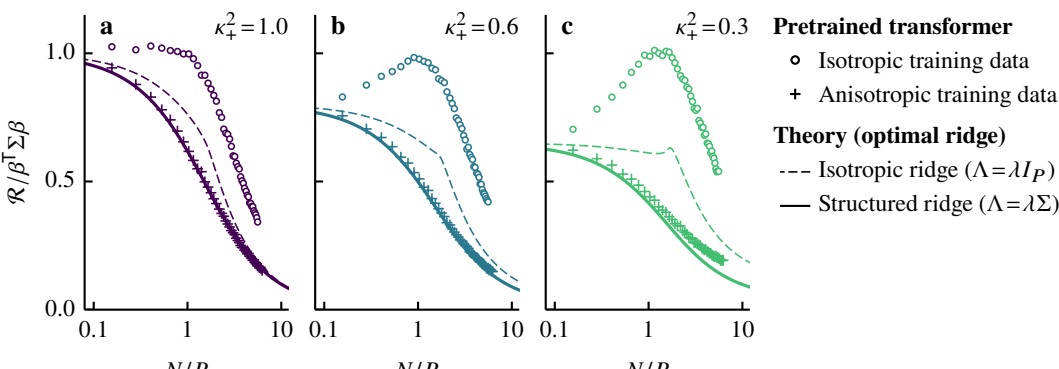

Figure 6: **Feature anisotropy modulates concept shift effects in transformer-based in-context regression.** We depict prediction risk on two-scale, anisotropic data (see §3.2) for optimal ridge regression (curves) and transformers (symbols) trained to solve linear regression tasks using in-context examples. The theoretical results are for optimally tuned isotropic and structured ridge penalties (dashed and solid curves, respectively). The transformers' results are for isotropic and anisotropic pretraining data (circles and plus symbols, respectively). Concept shift affects only high-variance features via the scaling factor $\kappa_+^2 = 1.0, 0.6, 0.3$ in **a**, **b**, and **c**, respectively ($\kappa_- = \cos\theta_\pm = 1$ throughout). Transformers trained on anisotropic data (pluses) closely follow the solid curve, suggesting that they effectively implement a structured regularization that adapts to feature anisotropy—effectively isotropizing the features and averaging the impact of concept shift ($\kappa^2 = (\kappa_+^2 + \kappa_-^2)/2 = (1 + \kappa_+^2)/2$). Transformers trained on isotropic data (circles) exhibit nonmonotonic risk under concept shift, but with much more pronounced peaks compared to theoretical predictions (dashed). Here $s_-/s_+ = 0.1$, $\rho_+ = \rho_- = 1/2$, $\pi_+ = \pi_- = 1/2$ and SNR=1.

concept shift affects only high-variance features, transformers exhibit markedly different behaviors depending on their training data distribution. Transformers pretrained on anisotropic data closely align with theoretical predictions for optimally tuned ridge regression with structured penalties, indicating that they can apply different penalties to features based on their variances. This strategy is equivalent to whitening the features, resulting in an isotropized regression problem with an effective concept shift scale parameter $\kappa^2 = (1 + \kappa_+^2)/2$. In contrast, transformers trained on isotropic data struggle to adapt optimally to anisotropic test data, exhibiting more pronounced nonmonotonic risk curves than theoretically predicted for isotropic ridge penalties. This difference becomes particularly evident as concept shift strengthens (panel c), suggesting that the priors learned during pretraining significantly influence how transformers adapt to concept shift in features with different scales. These results highlight the importance of data structure in determining how models respond to distribution shifts and demonstrate that transformers can implicitly learn sophisticated regularization strategies when exposed to appropriately structured training data. We note that this agreement between theory and experiments is not trivial, as transformers perform in-context regression on finite data, in finite dimensions, and via a learned, data-driven algorithm, whereas our theoretical analyses are based on optimally tuned ridge regression in the thermodynamic limit.

# 5 Classification experiments

So far we have focused on regression problems. In this section, we experimentally test whether the key insights from our theory extend to the qualitatively different setting of classification, demonstrating that they likely capture more general phenomena of learning under concept shift. We consider standard multinomial logistic regression for MNIST [40] and FashionMNIST [41], using Adam optimizer with a minibatch size of 500 and a learning rate of 0.001 for 2,000 epochs.

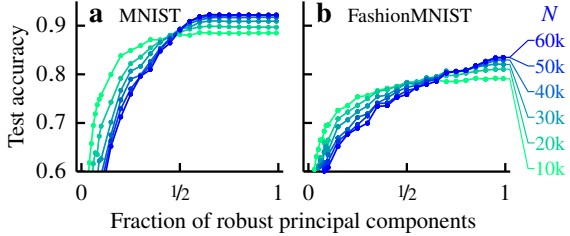

Figure 7: **Concept shift qualitatively changes data dependence of test accuracy** in (a) MNIST and (b) FashionMNIST experiments for various training data size $N$ (see legend). See §5 for details.

To vary training sample size $N$, we choose training data points at random (without replacement); all of the training data is used when $N = 60,000$.

We modify the test datasets by performing principal component analysis (PCA) on the images and designating the resulting features as either 'robust' or 'nonrobust' (cf. Fig 2c; see also §3.1). We shuffle nonrobust features across data points to decorrelate them from the labels while preserving marginal statistics, whereas robust features are unchanged. In Fig 7, we let lower-variance features be nonrobust and parametrize concept shift strength with the variance threshold that separates robust from nonrobust features. We depict the test accuracy as a function of concept shift strength, and observe that when concept shift is strong and few features are robust, more data hurts test accuracy, in qualitative agreement with our results for linear regression problems (cf. Figs 2 and 5). The qualitative agreement between our theoretical results and experiments on transformers, MNIST, and FashionMNIST suggests that the phenomenon we identify is not specific to linear regression, providing further validation for the broader relevance of our theory.

## 6 Related works

There is a significant body of work on out-of-distribution generalization with covariate shift, i.e., domain generalization (see, e.g., Refs [8, 42]), but significantly less work has been done on concept shift generalization. However, even within covariate shift, an understanding of out-of-distribution generalization remains elusive [11]. Within concept shift, work has tended to focus instead on detection and mitigation strategies rather than generalization (see, e.g., Ref [29]). Our work contributes to a distinct paradigm. We use tools from statistical mechanics and random matrix theory to derive an exact, analytical characterization of generalization behavior under a discrete shift. This approach allows us to uncover new phenomena, such as the sharp phase transition between weak and strong concept shift regimes.

Linear regression has proved a particularly useful setting for investigating learning phenomena. Random matrix universality makes it well-suited for theoretical investigations, yielding much-needed insights into high-dimensional learning [18, 19, 21–25, 43], including covariate shift generalization [44]. Linear models have also contributed substantially to our developing understanding of in-context learning in transformers [31–33, 35–38]; our work leverages this setting for experimental validation of our theory. Other works use linear attention mechanisms to analyze transformer models (see, e.g., Ref [45]).

The nonmonotonic behavior we identify is distinct from double descent phenomena [26, 27]. Indeed, our theoretical results in Figs 2-4 are for optimal regularization, for which double descent is absent [17, 18, 28], thus demonstrating that concept shift induces risk nonmonotonicity through fundamentally different mechanisms.

We build on analytical techniques developed for high-dimensional ridge regression, but address a phenomenon distinct from prior works. In particular, optimally tuned models can exhibit nonmonotonic risk due to model misspecification [18]—i.e., a mismatch between model class and data-generating process, irrespective of test-time shift. This mechanism differs from distribution shift, in which the data-generating process itself changes at test time. Our framework specifically isolates the effects of concept shift in correctly specified models.

In-context learning of transformers has been studied under covariate shift [36, 46, 47] but not with respect to concept shift. Song *et al.* [48] studied how transformers generalize to out-of-distribution tasks through symbolic manipulation via induction heads, but the scenario of within-context concept shift remains underexplored. While Agarwal *et al.* [34] showed that longer context windows typically benefit in-context learning, they also observed performance degradation in tasks such as MATH.

## 7 Conclusion and outlook

We introduce a ridge regression model for concept shift. Our model is exactly solvable in the high dimensional limit. We show that concept shift can change the qualitative behavior of generalization performance; for sufficiently strong concept shift, ridge regression fails to generalize even with infinite data (Figs 2 and 3). In addition, we demonstrate that input anisotropy can lead to nonmonotonic data dependence of prediction risk. In particular, too much data can harm generalization (Fig 4a and b) and

more data may only improve generalization above a certain threshold (Fig 4c and d). We emphasize that our results are for optimally tuned ridge regression and thus differ from risk nonmonotonicity due to double and multiple descent phenomena which are absent under optimal regularization [28].

Taken together, our work provides a fresh perspective on a lesser-studied mode of distribution shift. Our model offers a relatively simple setting for building intuitions and testing hypotheses about concept shift generalization. Although our theoretical analyses are exact only for ridge regression in the high-dimensional limit, the qualitative conclusions generalize beyond this idealized setting. Indeed, our theoretical prediction agrees *quantitatively* with experiments on regression in finite dimensions from finite samples, using the in-context learning ability of a transformer model as a learning algorithm. Additionally, our classification experiments suggest that the insights from our theory may apply in more general settings. Finally, our theoretical results for anisotropic features have implications for understanding data dimension reduction techniques—such as principal component regression in which low-variance features are discarded—under concept shift. While we acknowledge that real-world concept shifts occur in more complex contexts, our approach allows us to isolate and precisely characterize key phenomena, establishing a theoretical foundation in tractable models and providing a critical step toward understanding more complex settings.

Our work has limitations that open avenues for future research. First, our theoretical results are derived for linear models in the high-dimensional limit and may not quantitatively hold for complex nonlinear systems. Second, we analyze a discrete concept shift, leaving the study of gradual or continuous shifts for future work. Finally, our work characterizes a failure mode but does not propose an algorithm for mitigating its effects. We hope our work provides a foundation for addressing these important questions and encourages further systematic investigations of the rich learning phenomena induced by concept shift

## Acknowledgments and Disclosure of Funding

Alex Nguyen is supported by NIH grant RF1MH125318. DJS was partially supported by a Simons Fellowship in the MMLS, a Sloan Fellowship, and the National Science Foundation, through the Center for the Physics of Biological Function (PHY-1734030). VN acknowledges research funds from the University of Sydney. This work was supported in part by the National Science Foundation and by DoD OUSD (R&E) under Cooperative Agreement PHY-2229929 (The NSF AI Institute for Artificial and Natural Intelligence).

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

## A Prediction risk in the thermodynamic limit

To derive prediction risk in the thermodynamic limit, we rewrite the nonasymptotic bias and variance contributions [Eq (5)] in terms of the resolvent operator $(\Psi + \lambda I_P)^{-1}$,

$$B(X) = (\beta - \tilde{\beta})^{\mathsf{T}} \tilde{\Sigma} (\beta - \tilde{\beta}) - 2\lambda \operatorname{Tr}\left(\beta(\beta - \tilde{\beta})^{\mathsf{T}} \tilde{\Sigma} \tfrac{1}{\Psi + \lambda I_P}\right) + \lambda^2 \operatorname{Tr}\left(\beta\beta^{\mathsf{T}} \tfrac{1}{\Psi + \lambda I_P} \tilde{\Sigma} \tfrac{1}{\Psi + \lambda I_P}\right) \tag{12}$$

$$V(X) = \sigma_{\tilde{\xi}}^2 \left[\tfrac{1}{N} \operatorname{Tr}\left(\tilde{\Sigma} \tfrac{1}{\Psi + \lambda I_P}\right) - \lambda \tfrac{1}{N} \operatorname{Tr}\left(\tilde{\Sigma} \tfrac{1}{(\Psi + \lambda I_P)^2}\right)\right]. \tag{13}$$

In the thermodynamic limit—$N, P \to \infty$ and $P/N \to \gamma \in (0, \infty)$—the above traces become deterministic (see Appendix B), and the bias and variance converge to (see also Patil *et al.* [30])

$$B(X) \to \mathcal{B} = (\beta - \tilde{\beta})^{\mathsf{T}} \tilde{\Sigma} (\beta - \tilde{\beta}) - 2\lambda \beta^{\mathsf{T}} \hat{G}_\Sigma(-\lambda) \tilde{\Sigma} (\beta - \tilde{\beta}) + \lambda^2 \nu'(-\lambda) \frac{\beta^{\mathsf{T}} \hat{G}_\Sigma(-\lambda) \tilde{\Sigma} \hat{G}_\Sigma(-\lambda) \beta}{\frac{1}{P} \operatorname{Tr}[\Sigma \hat{G}_\Sigma^2(-\lambda)]}$$

$$+ \lambda^2 \gamma m(-\lambda)^2 \nu'(-\lambda) \frac{\beta^{\mathsf{T}} \hat{G}_\Sigma(-\lambda) \left(\operatorname{Tr}[\tilde{\Sigma}\Sigma \hat{G}_\Sigma^2(-\lambda)]\Sigma - \operatorname{Tr}[\Sigma^2 \hat{G}_\Sigma^2(-\lambda)]\tilde{\Sigma}\right) \hat{G}_\Sigma(-\lambda) \beta}{\operatorname{Tr}[\Sigma \hat{G}_\Sigma^2(-\lambda)]} \tag{14}$$

$$V(X) \to \mathcal{V} = \sigma_{\tilde{\xi}}^2 \gamma \left(\nu(-\lambda) \frac{\operatorname{Tr}[\tilde{\Sigma}\hat{G}_\Sigma(-\lambda)]}{\operatorname{Tr}[\Sigma\hat{G}_\Sigma(-\lambda)]} - \lambda \nu'(-\lambda) \frac{\operatorname{Tr}[\tilde{\Sigma}\hat{G}_\Sigma'(-\lambda)]}{\operatorname{Tr}[\Sigma\hat{G}_\Sigma'(-\lambda)]}\right), \tag{15}$$

where $\hat{G}_\Sigma(z) \equiv \frac{1}{m(z)\Sigma - z I_P}$, $m(z) \equiv \frac{1}{1 + \gamma \nu(z)}$ and $\nu(z) = \frac{1}{P} \operatorname{Tr}[\Sigma \hat{G}_\Sigma(z)]$ with $\nu(z) \in C^+$. We note that the last term of the bias vanishes for $\Sigma = \tilde{\Sigma}$, and covariate shifts affect the variance term but concept shift does not.

## B Spectral convergence for random matrix traces

Let $\Psi$, $\Theta$ and $A$ denote $P \times P$ matrices, $I_P$ the identity matrix in $P$ dimensions and $z$ a complex scalar outside the positive real line. Assume the following: (i) $A \in \mathbb{R}^{P \times P}$ is symmetric and nonnegative definite, (ii) $\Theta \in \mathbb{R}^{P \times P}$ has a bounded trace norm $\operatorname{Tr}[(\Theta^{\mathsf{T}}\Theta)^{1/2}] \in [0, \infty)$ and (iii) $\Psi = \frac{1}{N}\Sigma^{1/2} Z Z^{\mathsf{T}} \Sigma^{1/2}$ where the entries of $Z \in \mathbb{R}^{P \times N}$ are iid random variables with zero mean, unit variance and finite $8 + \varepsilon$ moment for some $\varepsilon > 0$, and $\Sigma \in \mathbb{R}^{P \times P}$ is a covariance matrix. In the limit $P, N \to \infty$ and $P/N \to \gamma \in (0, \infty)$, we have [49]

$$\operatorname{Tr}\left(\Theta \frac{1}{\Psi + A - z I_P}\right) \to \operatorname{Tr}\left(\Theta \frac{1}{\frac{1}{1 + \gamma c(z; A)}\Sigma + A - z I_P}\right), \tag{16}$$

where $c_A(z) \in \mathbb{C}^+$ is the unique solution of

$$c(z; A) = \frac{1}{P} \operatorname{Tr}\left(\Sigma \frac{1}{\frac{1}{1 + \gamma c(z; A)}\Sigma + A - z I_P}\right). \tag{17}$$

First we consider the trace of terms linear in the resolvent $(\Psi - z I_P)^{-1}$, appearing in Eqs (12) & (13). Setting $A = 0$ in Eq (16) gives

$$\operatorname{Tr}\left(\Theta \frac{1}{\Psi - z I_P}\right) \to \operatorname{Tr}\left(\Theta \frac{1}{\frac{1}{1 + \gamma \nu(z)}\Sigma - z I_P}\right) \tag{18}$$

where $\nu(z) \equiv c(z; 0)$ is the solution of

$$\nu(z) = \frac{1}{P} \operatorname{Tr}\left(\Sigma \frac{1}{\frac{1}{1 + \gamma \nu(z)}\Sigma - z I_P}\right) \quad \text{with} \quad \nu(z) \in \mathbb{C}^+. \tag{19}$$

In general, this self-consistent equation does not have a closed-form solution. One exception is the isotropic case $\Sigma = I_P$ for which

$$\nu_{\Sigma = I_P}(z) = \frac{1}{2\gamma z}\left[1 - \gamma - z - \sqrt{(1 - \gamma - z)^2 - 4\gamma z}\right]. \tag{20}$$

Next we obtain the asymptotic expression for the trace of terms quadratic in the resolvent, such as those in Eqs (12) & (13). We let $A = \mu B$ with $\mu > 0$. Differentiating Eq (16) with respect to $\mu$ and taking the limit $\mu \rightarrow 0^+$ yields

$$\text{Tr}\left(\Theta \frac{1}{\Psi - zI_P} B \frac{1}{\Psi - zI_P}\right) \rightarrow \text{Tr}\left(\Theta \frac{1}{\frac{1}{1+\gamma\nu(z)}\Sigma - zI_P}(d(z;B)\Sigma + B)\frac{1}{\frac{1}{1+\gamma\nu(z)}\Sigma - zI_P}\right). \tag{21}$$

Here we define

$$d(z;B) \equiv \left.\frac{d}{d\mu}\frac{1}{1 + \gamma c(z;\mu B)}\right|_{\mu \rightarrow 0^+} \tag{22}$$

$$= \frac{\gamma\frac{1}{P}\text{Tr}\left(B\frac{\Sigma}{(\Sigma-(1+\gamma\nu(z))zI_P)^2}\right)}{1 - \gamma\frac{1}{P}\text{Tr}[(\frac{\Sigma}{\Sigma-(1+\gamma\nu(z))zI_P})^2]} \tag{23}$$

$$= \frac{\gamma\nu'(z)}{(1 + \gamma\nu(z))^2}\frac{\frac{1}{P}\text{Tr}\left(B\frac{\Sigma}{(\frac{1}{1+\gamma\nu(z)}\Sigma-zI_P)^2}\right)}{\frac{1}{P}\text{Tr}\frac{\Sigma}{(\frac{1}{1+\gamma\nu(z)}\Sigma-z)^2}}. \tag{24}$$

where the last equality follows from

$$\nu'(z) = \frac{\frac{1}{P}\text{Tr}\frac{\Sigma}{(\frac{1}{1+\gamma\nu(z)}\Sigma-z)^2}}{1 - \gamma\frac{1}{P}\text{Tr}[(\frac{\Sigma}{\Sigma-(1+\gamma\nu(z))zI_P})^2]}. \tag{25}$$

For $B = I_P$, the spectral function $d(z;B)$ reduces to

$$d(z;I_P) = \frac{\gamma\nu'(z)}{(1 + \gamma\nu(z))^2}. \tag{26}$$

Our result for prediction risk in the thermodynamic limit is based on the asymptotic traces in Eqs (18) & (21).

