# OpenReview forum: "Generalization vs Specialization under Concept Shift"
_NeurIPS.cc/2025/Conference — NeurIPS 2025 poster_

### Official Review · Reviewer_hzmg · 2025-06-22

**Clarity:** 2
**Significance:** 2
**Originality:** 3
**Rating:** 4
**Confidence:** 3

**Summary:**

The paper studies regression/classification performance with ridge regression under concept shift and show that concept shift can change the qualitative behavior of generalization behavior. Authors report experimental results for regression tasks using pretrained transformers to solve linear regression and for classification tasks using well known MNIST and FashionMNIST datasets. Empirical investigations show that ridge regression fails to generalize even with infinite data under a strong enough concept shift. They also demonstrate that too much data can harm generalization when dealing with anisotropy in features.

**Questions:**

- How can researchers and machine learning practitioners leverage the learning from this work in their work? In particular, detecting or predicting concept shift in test data under real world settings can be hard or impossible which makes it challenging whether or not to consider that takeaways from this work at training stage.

**Ethical Concerns:**

["NO or VERY MINOR ethics concerns only"]

**Limitations:**

- Please refer to the first two bullets under the **weaknesses** section of this review.
- The high-dimensional limit assumption made in the paper does not necessarily match the practical settings.

**Paper Formatting Concerns:**

No major formatting concerns.

**Quality:**

3

**Strengths And Weaknesses:**

**Strengths**
- The topic addressed by this paper is of great significance since concept shift remains an active challenge for many ML systems in practice.
- The theoretical aspect of the paper is solid and it provides a rigorous understanding of model performance under concept shift. The studied setting, i.e. concept shift, is well defined.
- Experimental results shed light into less explored aspect of generalization performance under the studied settings e.g. empirical studies of how anisotropic features can cause more data hurting generalization performance.

**Weaknesses**
- Authors study concept shift's impact on ridge regression, an inherently linear model, which may make the observed model behavior specific to this oversimplified settings.
- The impact of concept shift on generalization performance is not well studied for classification tasks since the studied datasets for classification are rather simple and do not pose real world concept shift challenges.
- The paper is not easy to read. e.g. instead of using intuitive and easy to understand X and Y axis titles, authors have decided to use mathematical statements. e.g. Figure captions do not highlight the main observations/takeaways.
- Limitations of the work are not clearly and comprehensively listed by the authors.

---

> ### Author Rebuttal · Authors · 2025-07-31
>
> Thank you for your constructive feedback and for highlighting the significance of our work. We appreciate the opportunity to clarify our methodology and improve our presentation.
>
> &nbsp;
>
> > ...ridge regression, an inherently linear model, which may make the observed model behavior specific to this oversimplified settings.
>
> > The high-dimensional limit assumption made in the paper does not necessarily match the practical settings.
>
> Thank you for raising these important points about our methodology. We agree that our theoretical settings are simple. However, this simplicity is precisely what enables rigorous theoretical analyses and controlled experiments that are required for developing an understanding of complex learning phenomena.
>
> In particular, _linear models_ have proved crucial in advancing our understanding of learning dynamics of deep networks (see, eg, Saxe et al, ICLR 2014; Lampinen & Ganguli, ICLR 2019; Ji & Telgarsky, ICLR 2019; Arora et al, ICLR 2019, Lin et al, NeurIPS 2024; Marion & Chizat, NeurIPS 2024; Braun et al, ICML 2025; Nam et al, ICML 2025). _Linear regression_ has provided a crucial, theoretically tractable setting for developing insights into emergent abilities of LLMs, such as in-context learning (see, eg, Garg et al, NeurIPS 2022; Raventos et al, NeurIPS 2023; Goddard et al, ICML 2025; Lu et al, PNAS 2025). _Ridge regression in high dimensions_ has repeatedly yielded unexpected insights into machine learning behaviors such as double descent and benign overfitting (see, eg, Bartlett et al, PNAS 2020; Wu & Xu, NeurIPS 2020; Hastie et al, Ann Stat 2022; Richards et al, AISTATS 2021; Mel & Ganguli, ICML 2021; Patil et al, ICML 2024). Furthermore, the study of _high-dimensional limits_ has led to powerful frameworks such as the Neural Tangent Kernel (NTK; Jacot et al, NeurIPS 2018), and the Neural Network and Gaussian Process correspondence (NNGP; Lee et al, ICLR 2018).
>
> Our work follows in this fruitful tradition, providing a solvable framework that reveals nontrivial behaviors under concept shift—such as nonmonotonic data dependence of prediction risk and sharp transitions between weak and strong concept shift regimes. These effects would be difficult, if not impossible, to isolate in more complex models.
>
> Crucially, our experiments are specifically designed to test whether these insights extend beyond this idealized setting. The strong agreement between our theory and our transformers and classification experiments suggests the phenomena we identify are not mere artifacts of our theoretical assumptions but hint at more universal principles.
>
> &nbsp;
>
> > The impact of concept shift on generalization performance is not well studied for classification tasks…
>
> This is an excellent point and an important direction for future research. Our work provides the first theoretical characterization of concept shift in a regression setting, and we agree that a similar, rigorous analysis for classification is an important next step. Indeed, our experiments on MNIST and FashionMNIST, which demonstrate that more data can hurt test accuracy under strong concept shift, provide compelling preliminary evidence that similar phenomena exist in classification problems. We hope our work inspires future theoretical investigation into this area.
>
> &nbsp;
>
> > The paper is not easy to read…
>
> Thank you for this valuable feedback on readability. We agree that we can make our figures more accessible without sacrificing technical precision. In our revised manuscript, we will:
>
> 1.  Expand all figure captions to explicitly state the key takeaway of each plot in plain language.
> 2.  Add descriptive text to the axis labels of our plots to supplement the mathematical notation.
>
> We believe these changes will significantly improve the clarity and impact of our results for a broader audience.
>
> &nbsp;
>
> > Limitations of the work are not clearly and comprehensively listed…
>
> Thank you for this important suggestion. We identify three primary limitations, which also represent exciting directions for future work.
>
> 1.  Our theoretical results are derived for linear models in the idealized limit and may not _quantitatively_ hold for more complex nonlinear systems and in practically relevant limits.
> 2.  We analyze discrete concept shift, leaving the study of gradual or continuous shifts for future work.
> 3.  Our work characterizes a failure mode but does not propose a new solution or algorithm for mitigating its effects.
>
> We will add a dedicated _Limitations and Future Work_ section to our revised manuscript to further clarify these points.
>
> &nbsp;
>
> > How can researchers and machine learning practitioners leverage the learning from this work in their work?
>
> Thank you for pointing out the opportunity to discuss the implications of our research.
>
> The main takeaway for practitioners and researchers is a new perspective on the modern understanding of generalization. The double descent phenomenon challenged the classical intuition that _more data is always better._ While subsequent work established that optimal regularization can mitigate double descent, our paper identifies a distinct setting where this principle is insufficient.
>
> We show that even with optimal regularization, concept shift can create a new regime where additional data is detrimental to generalization performance. For researchers, our work provides a novel, tractable theoretical framework and identifies a rich set of phenomena (eg, sharp transitions and risk nonmonotonicity) worthy of further investigation.

---

> > ### Comment · Reviewer_hzmg · 2025-08-06
> >
> > I'd like to thank the authors for their detailed rebuttal.
> >
> > - Authors' explanations about simplicity of theoretical settings enables rigorous analyses makes sense. I hope that either the authors or other researchers in the future build up on this body of work and study more complex/non-linear settings.
> >
> > - Thanks for your willingness to address my comments about readability and limitation is greatly appreciated.
> >
> > - Finally, thanks for explaining the takeaways for practitioners and researchers.
> >
> > After going through the rebuttals and other reviewers' comments, I would like to keep my overall rating.

---

### Official Review · Reviewer_y7QV · 2025-07-01

**Clarity:** 3
**Significance:** 2
**Originality:** 3
**Rating:** 4
**Confidence:** 2

**Summary:**

The paper develops an exact theory of concept shift scenario in which the relationship between inputs and labels changes at test time, and studies how this shift alters the generalization behaviour of learning algorithms. They verify the analytic predictions with experiments, finding close quantitative and qualitative agreement. The paper provides a detailed theoretical and experimental characterization of how concept shift affects generalization.

**Questions:**

1.	The paper focuses on ridge regression. How do the findings extend to other machine learning models?
2.	The experiments are limited to MNIST and FashionMNIST. Could more diverse datasets be used to validate the findings? How do the results vary across different types of data distributions and problem domains?
3.	What advantages does this work offer over previous solutions?

**Ethical Concerns:**

["NO or VERY MINOR ethics concerns only"]

**Final Justification:**

Thank you for your detailed response to my review. I appreciate the time and effort you have put into addressing the comments. I have read your rebuttal carefully and have no further questions. After consideration, I will be maintaining my original score. I wish you the best with your submission.

**Limitations:**

yes

**Quality:**

3

**Strengths And Weaknesses:**

## Srengths:
1.	The paper delves deep into the issue of machine learning models under concept shift, centering on ridge regression to derive exact expressions. The methodology is rigorous.
2.	The paper is well-structured and logically coherent. It starts with the problem of distribution shift, narrows down to concept shift, and elaborates on the research background and significance. It then proceeds to theoretically analyze ridge regression under concept shift. The experimental section follows naturally.
3.	Understanding the generalization of machine learning models under distribution shift is crucial for developing safe and robust techniques. This paper focuses on the relatively less-studied concept shift, offering valuable insights.
## Weaknesses
1.	While the paper provides a detailed analysis of concept shift under ridge regression, the scope of its research is relatively limited.
2.	Some mathematical derivations and notations in the paper are quite complex, which may pose challenges for readers without a strong mathematical background.
3.	The practical applicability of the findings remains to be further demonstrated.

---

> ### Author Rebuttal · Authors · 2025-07-31
>
> Thank you for highlighting the significance and rigor of our work. We appreciate the opportunity to clarify our contributions and the scope of our findings. Please see below our point-by-point response to specific comments with quotes rearranged for clarity.
>
> &nbsp;
>
> > …the scope of its research is relatively limited.
>
> Thank you for pointing out the opportunity to articulate our contributions.
>
> We emphasize that our main contribution is in deriving an exact, analyzable framework for characterizing concept shift and its effects on generalization properties. We view our work as establishing an important theoretical foundation upon which future research can study other models of concept shift in different settings. The mathematical clarity achieved through our approach enables precise characterization of phenomena that would be difficult to quantify in more complex settings.
>
> To the best of our knowledge, we are the first to:
> *  develop a theoretical framework that isolates the effects of concept shift—an important and relatively less studied mode of distribution shift;
> *  explain precisely why and how more training data can hurt generalization performance under concept shift, illustrating its effects in several specific settings, including coefficient shrinking, rotation, and feature robustness;
> *  identify and characterize a sharp transition that separates weak and strong concept shift regimes;
> *  show that feature anisotropy creates qualitatively different patterns of risk nonmonotonicity depending on whether concept shift affects high or low-variance features.
>
> We will revise our Introduction to emphasize these points.
>
> &nbsp;
>
> > The practical applicability of the findings remains to be further demonstrated
>
> > What advantages does this work offer over previous solutions?
>
> Thank you for raising this important point. Other work focuses primarily on detecting the presence of a concept shift. In contrast, our work does not propose a new solution or algorithm to detect or mitigate concept shift. Instead, we provide a fundamental characterization of model failure under concept shift, revealing counterintuitive phenomena and laying the foundation for the future development of robust solutions.
>
> &nbsp;
>
> > The paper focuses on ridge regression. How do the findings extend to other machine learning models?
>
> We appreciate the opportunity to highlight our experiments. While our theory is derived for ridge regression, we provide experimental evidence that the key insights extend to a qualitatively different learning problem. In classification settings, our MNIST and FashionMNIST experiments demonstrate that under a strong concept shift, test accuracy can decrease with more training data. This result strongly suggests that the phenomenon we identify is not limited to regression models, thus providing convincing validation for the broader relevance of our theory.
>
> &nbsp;
>
> > The experiments are limited to MNIST and FashionMNIST. Could more diverse datasets be used to validate the findings? How do the results vary across different types of data distributions and problem domains?
>
> This is an excellent point. We address this with our transformer experiments, which test our theory in a modern setting. The agreement between our theory and this complex model is not trivial. Our theoretical analyses are based on optimally tuned ridge regression in the thermodynamic limit. In contrast, transformers perform in-context regression on finite data (ie, their context) in finite dimensions via a learned, data-driven algorithm (as opposed to hardcoded ridge regression). Our transformer experiments thus serve as validation for our theoretical results by demonstrating that the phenomena we identify extend beyond idealized settings. In particular, we show that similar behavior—such as a transition between weak and strong concept shift regimes—emerges in a data-driven learning algorithm without (a) explicitly implementing ridge regression, (b) working at infinite dimensions, or (c) manually tuning regularization.
>
> These results validate our analysis method of studying tractable concept shift models where explicit expressions can be derived and demonstrate that studying these models can reveal qualitative generalization behaviors that are also observed in large-scale data-driven learning. Together, our experimental results on transformer models and classification tasks hint at universal aspects of learning under concept shift that our theoretical model effectively captures. This highlights the value of our theory as a minimal model for investigating concept shift and its effects on generalization.
>
> We will revise our experiment section to better motivate these experiments and discuss how they provide a nontrivial validation of our theoretical results.
>
> &nbsp;
>
> > …mathematical derivations and notations in the paper are quite complex…
>
> We appreciate this feedback on the accessibility of our paper. While the mathematical details are essential for the rigor of our analysis, we agree that the high-level insights should be as clear as possible. In our revised manuscript, we will edit the Abstract, Introduction, and Conclusion sections to better distill the key takeaways and their implications in more accessible language, without diluting the core technical results.

---

> > ### Comment · Reviewer_y7QV · 2025-08-04
> >
> > Thank you for your detailed response to my review. I appreciate the time and effort you have put into addressing the comments. I have read your rebuttal carefully and have no further questions. After consideration, I will be maintaining my original score. I wish you the best with your submission.

---

### Official Review · Reviewer_fv8w · 2025-07-02

**Clarity:** 2
**Significance:** 2
**Originality:** 3
**Rating:** 4
**Confidence:** 3

**Summary:**

This paper presents a theoretical study of how concept shift—a change in the input-label relationship—affects generalization in ridge regression. The authors provide exact expressions for prediction risk in the high-dimensional limit and identify a sharp phase transition: when concept shift is strong, adding more training data can actually worsen test performance. Experiments using in-context learning with transformers and classification on MNIST/FashionMNIST support the theory.

**Questions:**

See the Strengths And Weaknesses section.

**Ethical Concerns:**

["NO or VERY MINOR ethics concerns only"]

**Final Justification:**

The authors have give a detaield response to clarify the coontributions of the paper and most of my concerns have been addressed.

**Limitations:**

yes

**Paper Formatting Concerns:**

no formatting concerns.

**Quality:**

2

**Strengths And Weaknesses:**

Strengths:
1. The paper identifies a clear and interesting phenomenon: more training data can hurt generalization under strong concept shift.
2. The theoretical analysis is rigorous and based on a solvable model, offering precise characterizations.
3. Experimental results on transformers and classification tasks are generally consistent with the theoretical predictions.

Weaknesses:
1. The theoretical results rely heavily on idealized settings: linear models, Gaussian features, and optimally tuned ridge regression. These are useful for analysis but may not reflect practical machine learning scenarios. Would similar effects appear with nonlinear models, neural networks, or non-Gaussian data?
2. The shift is modeled only as scaling or rotating the regression coefficients. In real-world problems, concept shift is often more complex, involving sparse, structured, or gradual changes. Discuss how realistic this modeling is, and whether the findings generalize to more practical types of shift.
3. While the paper highlights its novelty, it does not sufficiently compare with prior theoretical work on concept drift, streaming learning, or temporal domain generalization.
4. The transformer experiments focus only on simple synthetic tasks (linear regression) and PCA-modified MNIST. These setups are still far from real-world settings.

---

> ### Author Rebuttal · Authors · 2025-07-31
>
> Thank you for your thoughtful comments. We appreciate the opportunity to clarify the motivation, scope and contributions of our work.
>
> &nbsp;
>
> First, we emphasize that our main contribution is in deriving an exact, analyzable framework for characterizing concept shift and its effects on generalization properties. We view our work as establishing an important theoretical foundation upon which future research can study other models of concept shift in different settings. The analytical precision achieved through our approach enables precise characterization of phenomena, such as the transition between weak and strong concept shift regimes, that would be difficult to quantify in more complex settings.
>
> We will revise our manuscript to better highlight the broader implications of our findings. Specifically, we will include our arguments below in the Introduction and Related Work sections to more clearly articulate our main contributions and how our theoretical framework can inform understanding of concept shift in more complex settings.
>
> Please see below our point-by-point response to specific comments with quotes rearranged for clarity.
>
> &nbsp;
>
> > The theoretical results rely heavily on idealized settings: linear models…
>
> We agree that our theoretical settings are simple. However, this simplicity is precisely what enables rigorous theoretical analyses and controlled experiments that are required for developing an understanding of complex learning phenomena.
>
> In particular, _linear models_ have proved crucial in advancing our understanding of learning dynamics of deep networks (see, eg, Saxe et al, ICLR 2014; Lampinen & Ganguli, ICLR 2019; Ji & Telgarsky, ICLR 2019; Arora et al, ICLR 2019, Lin et al, NeurIPS 2024; Marion & Chizat, NeurIPS 2024; Braun et al, ICML 2025; Nam et al, ICML 2025). _Linear regression_ has provided a crucial, theoretically tractable setting for developing insights into emergent abilities of LLMs, such as in-context learning (see, eg, Garg et al, NeurIPS 2022; Raventos et al, NeurIPS 2023; Goddard et al, ICML 2025; Lu et al, PNAS 2025). _Ridge regression in high dimensions_ has repeatedly yielded unexpected insights into machine learning behaviors such as double descent and benign overfitting (see, eg, Bartlett et al, PNAS 2020; Wu & Xu, NeurIPS 2020; Hastie et al, Ann Stat 2022; Richards et al, AISTATS 2021; Mel & Ganguli, ICML 2021; Patil et al, ICML 2024). Our work follows in this fruitful tradition.
>
> Our work provides a solvable framework that reveals nontrivial behaviors under concept shift—such as nonmonotonic data dependence of prediction risk and phase transitions between weak and strong concept shift regimes. These effects would be difficult, if not impossible, to isolate in more complex models.
>
> &nbsp;
>
> > ...simple synthetic tasks (linear regression) and PCA-modified MNIST [...] are still far from real-world settings.
>
> We do not design our experiments to solve a complex real-world task, but to validate the insights from our theory in settings beyond which we can access analytically. We show that the key phenomenon—the weak-to-strong concept shift transition—persists in more complex settings, including transformer in-context learning and classification tasks on image datasets.
>
> The strong agreement between our theory and these experiments hints at universal aspects of learning under concept shift that our theoretical model effectively captures. This highlights the value of our theory as a principled minimal model for investigating concept shift and its effects on generalization.
>
> &nbsp;
>
> > The shift is modeled only as scaling or rotating the regression coefficients. In real-world problems, concept shift is often more complex…
>
> While we acknowledge that real-world concept shifts occur in more complex contexts, establishing a theoretical foundation in tractable models and verifying it in basic settings is a critical step toward understanding these phenomena. Crucially, for the linear data-generating process we study, these two primitives—changes in magnitude (scaling) and direction (rotation)—fully characterize any possible discrete concept shift. This approach allows us to isolate and precisely characterize key phenomena in a way that would be otherwise difficult.
>
> &nbsp;
>
> > ...does not sufficiently compare with prior theoretical work on concept drift, streaming learning, or temporal domain generalization.
>
> Thank you for this suggestion. Our work contributes to a distinct paradigm from much of the literature on concept drift and streaming learning.
>
> While many prior works focus on developing algorithms for _detecting_ and _adapting_ to drift, or on providing worst-case generalization bounds, we use tools from statistical mechanics and random matrix theory to derive an exact, analytical _characterization of generalization behavior_ under discrete concept shift. This approach allows us to uncover new phenomena, such as the sharp transition at the heart of our work. We will revise our Related Work section to better articulate this important distinction and provide representative references for other paradigms.

---

> > ### Comment · Reviewer_fv8w · 2025-08-03
> >
> > Thank you to the authors for thoroughly addressing my questions and for their efforts to clarify the contributions of the paper. After carefully reviewing the responses, I have raised the rating.

---

### Official Review · Reviewer_qGpA · 2025-07-02

**Clarity:** 3
**Significance:** 3
**Originality:** 3
**Rating:** 5
**Confidence:** 5

**Summary:**

The paper analyzes the mean-field behavior of generalization under **concept shift** in ridge regression, where the input–label mapping changes at test time. The authors derive a closed-form expression for the test error and study its properties on simple data models under *optimal* in-distribution regularization. They uncover a phase transition that separates a **generalization phase** (more data helps) from a **specialization phase** (more data hurts). These insights are then applied to in-context learning (ICL) and classification tasks, and validated in realistic experiments.

**Questions:**

See above

**Ethical Concerns:**

["NO or VERY MINOR ethics concerns only"]

**Final Justification:**

This paper theoretically studies the generalization of linear models under general concept shifts, which are relevant for modern transformers. Their results are clearly stated and supported by extensive numerical experiments.

**Limitations:**

The study is limited to simple Gaussian data.

**Quality:**

3

**Strengths And Weaknesses:**

### Strengths

1. The paper is well written and easy to follow; the problem setting and theoretical results are presented clearly.
2. Asymptotics of regression under concept shift are not well studied; the authors make a solid contribution by providing an intuitive mathematical exposition.
3. They identify a novel “double-descent” behavior that interestingly appears only at test time.
4. The theory is applied to ICL and shown to hold in non-trivial experimental settings.

### Weaknesses

The theory focuses on *optimal* ridge regression to isolate generalization behavior under concept shift. With sub-optimal ridge, in-distribution generalization always degrades, but it is unclear whether the same holds for the OOD case. Optimal ridge may overspecialize the predictor to the training task and thus potentially contribute to poorer OOD performance. Couldn't there be an alternative ridge parameter which marginally worsens ID risk yet substantially improve OOD risk?

Also, the previous literature mainly considered optimizing ridge parameter with respect to OOD generalization (ref.\[30,43]). How would that change the story? If optimally tuned ridge regression for ID yields an increasing OOD test risk, would tuning to optimal OOD ridge parameter have a similar effect on ID generalization?

Below are a few comments/suggestions

1. **Line 85.** The risk need not approach to zero, but rather to an irreducible error (see ref. \[43]) for generic feature maps. Likewise, the statement in line 88 is confusing; it would make more sense to me if $\tilde\beta_1 = \beta_1$ and $\tilde\beta_{i>1} = 0$. It is just a technicality, but it confused me the first time I read it.
2. **Line 117.** *nonrobut* → *non-robust*.
3. Please clarify how your results differ from those of **Yue et al. (2023)** ([https://openreview.net/forum?id=Jw3ck7FWZh](https://openreview.net/forum?id=Jw3ck7FWZh)).
4. What is the nature of the weak-to-strong concept-shift phase transition? Does it constitute a genuine phase transition in the strict statistical-mechanics sense—for example, does the risk exhibit a non-analyticity in κ cos θ—or is it merely a sharp crossover? A brief discussion would help readers interpret the result.

---

> ### Author Rebuttal · Authors · 2025-07-31
>
> Thank you for the positive and thorough review! We are grateful for your insightful questions and detailed feedback, which helped us improve our paper. Please see below our point-by-point response with quotes rearranged for clarity.
>
> &nbsp;
>
> > The theory focuses on optimal ridge regression to isolate generalization behavior under concept shift.
>
> We deliberately chose this setup to isolate the effects of concept shift from the confounding effects of double descent, another phenomenon where prediction risk can increase with more data (Belkin et al, PNAS 2019; Nakkiran et al, ICLR 2020). Focusing on optimal regularization, where double descent is absent, allows us to unambiguously demonstrate that concept shift is a distinct mechanism that can cause more data to hurt generalization performance.
>
> &nbsp;
>
> > Optimal ridge may overspecialize the predictor to the training task […] Couldn't there be an alternative ridge parameter which marginally worsens ID risk yet substantially improve OOD risk?
>
> > …optimizing ridge parameter with respect to OOD generalization […] How would that change the story? If optimally tuned ridge regression for ID yields an increasing OOD test risk, would tuning to optimal OOD ridge parameter have a similar effect on ID generalization?
>
> Thank you for this excellent question. Your intuition is correct. Indeed, tuning the regularization parameter λ away from the in-distribution (ID) optimal value, λ* = γ/snr, can improve out-of-distribution (OOD) performance. Concept shift with *κ*cos*θ* > 1 benefits from λ < λ* and *κ*cos*θ* < 1 from λ > λ*. Intuitively, the stronger concept shift is (ie, when *κ*cos*θ* is small or negative), the less you should trust the input features; improving OOD generalization thus requires a greater λ, which decreases the relative influence of training data on the learned predictor. This behavior reflects the fundamental competition between the inductive bias induced by ridge regularization and the signal from the data.
>
> This competition also underpins our key finding that a sharp transition exists between weak and strong concept shift regimes (Fig 3). Marginal deviations from λ* do not change the qualitative nature of this transition.
>
> While previous works considered OOD-optimal regularization, our focus on ID-optimal regularization is motivated by its practical relevance. Using optimal hyperparameters is a common practice, and optimizing regularization for unseen ID data is possible via methods such as cross-validation. Our transformer experiments, which are trained only on ID data, exemplify how it is possible to achieve virtually optimal ID generalization performance in practice.
>
> We have revised our manuscript to clarify this important point.
>
> &nbsp;
>
> >  how your results differ from those of Yue et al.
>
> Thank you for bringing this interesting work to our attention. We see our work as complementary to theirs, as both demonstrate the power of solvable models in generating new insights. However, we address fundamentally different questions.
>
> First, we focus on the phenomenon of concept shift within the framework of ridge regression, whereas Yue et al study how in-context learning with linear attention mechanisms can solve in-distribution linear regression tasks and don’t focus on concept shift. Second, our work uses ridge regression as a theoretically tractable model to derive exact, analytical results on concept shift. Our transformer experiments serve primarily as experimental validation, demonstrating that these insights generalize to practical architectures and optimization settings. In contrast, the transformer model in Yue et al is their primary object of study.
>
> We added a reference to and a discussion of this work to our Related Work section to better highlight these distinctions.
>
> &nbsp;
>
> > What is the nature of the weak-to-strong concept-shift phase transition? Does it constitute a genuine phase transition in the strict statistical-mechanics sense…
>
> Thank you for pointing out the opportunity to improve the clarity of our work.
>
> While the mappings between the transitions in learning problems and those separating phases of matter are instructive, they can be inexact, as many observables in the former do not have direct analogues in the latter and vice versa. In our work, the boundary between weak and strong concept shift is defined by an abrupt change in the utility of additional training data, from improving to degrading OOD generalization. This abrupt change manifests as a nonanalytic kink in the minimum OOD risk at the boundary (Fig 3), and this kink is sharp only in the strict thermodynamic limit. We have included this discussion in our updated paper.
>
> &nbsp;
>
> > Line 85. The risk need not approach to zero, but rather to an irreducible error
>
> You are correct that, in general, prediction risk includes an irreducible term due to intrinsic label noise. In our work, we follow the convention of Hastie et al (Ann Stat 2022) where the risk is the expected squared error between the predictor and the mean of the test-time conditional distribution (Eq 4 in our paper). This definition effectively subtracts the irreducible noise variance, allowing for a cleaner analysis of the error attributable to the learning algorithm. We have clarified this definition in our revised paper.
>
> &nbsp;
>
> > Line 117. nonrobut → non-robust.
>
> Thank you for catching this typo. We have corrected it.

---

> > ### Comment · Reviewer_qGpA · 2025-08-01
> > **Follow-up**
> >
> > I thank the authors for answering my questions and clarifying my concerns. Their response answered all my questions, and accordingly, I will increase my score. However, I also have a follow-up question regarding the generalization vs. specialization phase in relation to LLMs.
> >
> > Assuming these insights also transfer to modern LLMs, could we say that they operate in the *generalization phase* since they tend to show better OOD performance with more data, and what would the theory imply? Would it imply that the test cases so far were predominantly weak concept shifts from ID data? And if so, what predictions can you make about the structure of strong concept shift examples? More generally, can you comment on what your theory implies about the *no-free-lunch theorem* for out-of-context generalization?
> >
> > Also, as in Fig.4b, OOD generalization may improve until a critical data size, depending on the spectral properties of the data. Is it possible, for example, to predict whether OOD generalization in LLMs will eventually worsen with increasing data size?

---

> > > ### Author Response · Authors · 2025-08-05
> > >
> > > Thank you for the positive feedback and for increasing your score! We appreciate your follow-up questions.
> > >
> > > While our work can inform thinking about OOD generalization in general, we are not yet ready to make direct predictions about LLMs specifically based on our idealized model. OOD generalization behaviors and their underlying mechanisms in transformers can be much more diverse and complex than our exactly solvable setting, and our understanding of these systems is still developing (see, eg, Ahuja & Lopez-Paz, ES-FoMO Workshop @ ICML 2023; Yadlowsky et al, DistShift Workshop @ NeurIPS 2023; Goddard et al, ICML 2025; Cai et al, arXiv:2506.09251).
> > >
> > > &nbsp;
> > >
> > > > ...could we say that they operate in the *generalization phase*... [and] that the test cases so far were predominantly weak concept shifts?
> > >
> > > > ...what predictions can you make about the structure of strong concept shift examples?
> > >
> > > If we had to speculate, we believe that LLM OOD generalization is often compositional, and this  is a ripe area for follow-up work.
> > >
> > > Our work would suggest that to determine the strength of concept shift, we would want to compare the OOD performance of models trained on different dataset sizes. This approach might offer a glimpse of what language tasks constitute strong concept shift.
> > >
> > > &nbsp;
> > >
> > > > ...what your theory implies about the no-free-lunch theorem for out-of-context generalization?
> > >
> > > Great question. We in fact see our work as an illustration of this theorem—under strong concept shift, better ID generalization comes at the cost of poorer OOD generalization.
> > >
> > > &nbsp;
> > >
> > > > ...as in Fig.4b... Is it possible... to predict whether OOD generalization in LLMs will eventually worsen with increasing data size?
> > >
> > >
> > > It may be possible to make such a prediction, and the answer likely depends on the details of the particular training corpus and evaluation task. But we do not currently have all the necessary pieces in place. For example, it is unclear how to quantify the extent to which a test task is out-of-distribution relative to the pretraining corpus. We hope our work inspires future investigations in this direction (see, eg, Ye et al, arXiv:2507.09875).

---

### Decision · Program_Chairs · 2025-09-17

**Decision:**

Accept (poster)

**Comment:**

The paper studies generalization under label shift in a simplified model of ridge regression and in the limit of data dimension going to infinity.
The reviewers appreciate the theoretical study and while concerned about the over simplified setup, hope that this work could serve as a starting point for analysis of closer to reality setups.